# *Cdk9* regulates a promoter-proximal checkpoint to modulate RNA polymerase II elongation rate in fission yeast

Gregory T. Booth[1], Pabitra K. Parua[2], Miriam Sansó[2], Robert P. Fisher [2] & John T. Lis[1]

Post-translational modifications of the transcription elongation complex provide mechanisms to fine-tune gene expression, yet their specific impacts on RNA polymerase II regulation remain difficult to ascertain. Here, in *Schizosaccharomyces pombe*, we examine the role of *Cdk9*, and related *Mcs6/Cdk7* and *Lsk1/Cdk12* kinases, on transcription at base-pair resolution with Precision Run-On sequencing (PRO-seq). Within a minute of *Cdk9* inhibition, phosphorylation of Pol II-associated factor, *Spt5* is undetectable. The effects of *Cdk9* inhibition are more severe than inhibition of *Cdk7* and *Cdk12*, resulting in a shift of Pol II toward the transcription start site (TSS). A time course of *Cdk9* inhibition reveals that early transcribing Pol II can escape promoter-proximal regions, but with a severely reduced elongation rate of only ~400 bp/min. Our results in fission yeast suggest the existence of a conserved global regulatory checkpoint that requires *Cdk9* kinase activity.

[1] Department of Molecular Biology and Genetics, Cornell University, 107 Biotechnology Building, 526 Campus Road, Ithaca, NY 14853-2703, USA. [2] Department of Oncological Sciences, Icahn School of Medicine at Mount Sinai, New York, NY 10029, USA. Correspondence and requests for materials should be addressed to J.T.L. (email: johnlis@cornell.edu)

Multiple kinases modify RNA polymerase II (Pol II) and associated pausing and elongation factors to regulate Pol II transcription and pre-mRNA processing[1,2]. For example, the concerted action of Cdk7, Cdk9, and Cdk12 are required for the early transition of Pol II from an initiating to a productively elongating and RNA processing complex[3–6]. Cdk7, the kinase subunit of the TFIIH general transcription factor complex (Mcs6 in Schizosaccharomyces pombe, Cdk7 in humans, Kin28 in S. cerevisiae), phosphorylates the largest subunit of Pol II within the C-terminal domain (CTD) at the ser5 and ser7 positions of heptad amino acid sequence, YSPTSPS, which is repeated 29 times in S. pombe[7–9]. The modification of Pol II by Mcs6 is critical for recruitment of the 5′ RNA capping machinery and release of the mediator complex at the earliest stages of gene transcription[10–13]. Ser2 of the CTD heptad repeat is also phosphorylated as Pol II elongates further into the gene body[14], and both Cdk9 and Lsk1 (Cdk12 in humans, Ctk1 in S. cerevisiae) have been suggested to modify this residue, possibly to facilitate 3′ RNA processing[15].

Similar to the Pol II CTD, targets within elongation factors can also mediate essential regulation of transcription elongation. Spt4 and Spt5 comprise a highly conserved elongation factor (5,6-dichloro-1-β-D-ribofuranosylbenzimidazole (DRB)-sensitivity inducing factor, or DSIF in metazoans)[16]. Spt5 possesses an unstructured C-terminal repeat (CTR) domain, with specific residues actively targeted by Cdk9[17]. Moreover, the structure of eukaryotic Spt4/5 bound to transcribing RNA Pol II reveals contacts with upstream DNA and nascent RNA exiting Pol II[18], and depletions of Spt5 in fission yeast globally reduce levels of elongating Pol II[19]. Cdk9 (Bur1 in S. cerevisiae) positively impacts transcription elongation[20], and has been suggested to convert Spt5 from a negative to positive elongation factor in humans[21]. Such observations may reflect regulated recruitment of elongation-coupled factors, as association of capping enzymes and polymerase-associated factor 1 (Paf1) with the transcription complex are influenced by the phosphorylation status of Spt5[22–25].

In most metazoan systems, the phosphorylation of Spt5 and negative elongation factor (NELF) by positive transcription elongation factor b (P-TEFb, the Cdk9/cyclin T1 complex), is thought to be required for the release of elongating Pol II from a promoter-proximal pause site[26,27]. This Cdk9-regulated process (referred to hereafter as pausing) is now known to occur on the vast majority of genes in mammals[28], serving to regulate gene expression, or to ensure appropriate processing of RNA and/or maturation of the elongation complex[29]. In budding yeast, which lack homologs of all NELF subunits, such pausing has not been observed, yet perturbation of Cdk9 and Spt5 impair normal transcription by Pol II[20,30,31]. In contrast, pause-like distributions of Pol II have been found in the fission yeast S. pombe, where they depend on Spt4[32]. In addition, S. pombe Cdk9 interacts with the triphosphatase component of the 5′ RNA capping apparatus (Pct1), possibly to alleviate a Spt4-5-induced checkpoint that ensures proper pre-mRNA processing[17]. Still, such a checkpoint in yeast remains largely speculative, and the relation to Cdk9-regulated pausing in metazoans is unknown.

Investigations of the direct influence of kinase activity on the dynamics and regulation of transcription in vivo requires highly selective inhibitors, which can be difficult to obtain. Mutant kinases have been designed with exquisite specificity for bulky ATP analogs that selectively inhibit their activity[33]. The use of such analog-sensitive (AS) strains in fission yeast revealed correlated changes in global mRNA levels upon inhibition of Mcs6 or Cdk9 within minutes of inhibition[34]. Inhibitions of an AS TFIIH kinase resulted in a shift of Pol II levels toward the 5′ ends of genes in budding yeast after 1 h[35] and impaired pausing and termination in humans after a day of inhibition[3,36]. However, even within just 1 h, the primary effects of kinase inhibition on Pol II dynamics may be masked, especially in yeast where genes are short and transcribed in a matter of minutes[37].

In this study, to gain an insight into the impact of transcription complex-targeting kinases on Pol II elongation, we use AS kinase mutants in the fission yeast S. pombe to rapidly and selectively inhibit the activities of Cdk9, Mcs6, and Lsk1 individually or in combination. Using precision run-on sequencing analysis, we observe global changes in transcriptional dynamics within minutes of treatment, indicative of the general influence of these kinases on transcription by Pol II. While almost no instantaneous changes in Pol II distribution result from the loss of lsk1[as] activity, inhibition of cdk9[as] and/or mcs6[as] dramatically alter the transcription landscape within 5 min, despite non-overlapping substrates. Spt5 phosphorylation becomes undetectable after just 1 min of cdk9[as] inhibition; however, unlike metazoans, elongating Pol II is not trapped at the promoter-proximal pause site[28]. Instead, a fine-scale time course of cdk9[as] inhibition reveals that promoter-proximal Pol II has a severely compromised elongation rate, while downstream Pol II appears less affected. Our results support the existence of a Cdk9-dependent early elongation checkpoint in fission yeast, which may reflect a primitive form of Cdk9-regulated promoter-proximal pausing in metazoans.

## Results

**Cdk9 is not required for promoter-proximal pause escape.** Recently, we reported promoter-proximal pause-like distributions of Pol II on many genes in fission yeast[32], yet whether escape from such pausing is regulated through kinase activity is not known. In metazoans, kinase inhibitors, including DRB and flavopiridol (FP), which primarily inhibit Cdk9, have been instrumental in the discovery of promoter-proximal pausing as a major regulatory hurdle for most genes[4,26–28]. However, off-target effects of chemical inhibitors make selective ablation of Cdk9 activity in vivo nearly impossible[38].

To isolate the immediate impact of loss of Cdk9 activity on Pol II pausing and elongation at single-nucleotide resolution in fission yeast, we performed PRO-seq in an AS mutant strain, cdk9[as], which is vulnerable to inhibition by the addition of 3-MB-PP1[34,39]. Phosphorylation of Spt5 became undetectable after 1 min of drug addition in cdk9[as] (Fig. 1a), while bulk measurements of phosphorylation on Pol II CTD residues were largely unaffected, within the timeframe tested (Supplementary Fig. 1a). The near-immediate loss of Spt5 phosphorylation upon cdk9[as] inhibition implies an extremely rapid action of the small molecule 3-MB-PP1 and supports a model of active, competing Spt5 dephosphorylation[40]. Taking advantage of the ability to rapidly inhibit AS kinase activity, PRO-seq libraries were prepared in two biological replicates (Supplementary Fig. 1b) from cdk9[as] cells treated for 5 min with 10 µM 3-MB-PP1 (treated) or an equivalent volume of DMSO (untreated). As a control, we prepared PRO-seq libraries from equivalently treated wild-type (wt) cells. Comparison of untreated cdk9[as] profiles with wt profiles revealed no obvious differences (Supplementary Fig. 2a) and addition of 3-MB-PP1 to wt cells produced almost no transcriptional changes (Supplementary Fig. 2b), indicating that this system has little to no basal phenotype of the AS mutation or off-target drug effects. To evaluate the effect of Cdk9 on promoter-proximal pause-like distributions, genes were divided into quartiles based on their pausing index (PI), which measures the enrichment of Pol II in the promoter-proximal region (TSS to +100 nt) relative to the gene body ("+" indicates downstream and "−" indicates upstream). Although we observed striking increases in early gene body PRO-seq signal (TSS to +1 kb) resulting from

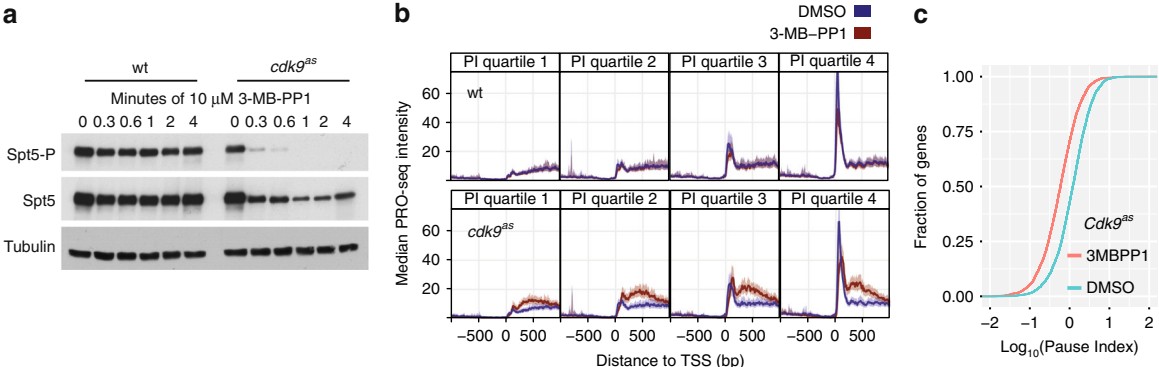

**Fig. 1** *Cdk9* is not required for promoter-proximal pause escape. **a** Western blot analysis using antibodies that specifically recognize pSpt5 compared with total *Spt5* in wt and *cdk9^as^* over a time course of cultures treated with 10 μM 3-MB-PP1. Total *Spt5* represents both phosphorylated and unphosphorylated forms. Tubulin serves as a loading control. **b** TSS-centered composite profiles of PRO-seq data before (blue) and after (red) 5 min of treatment with 10 μM 3-MB-PP1 for genes grouped by quartiles of increasing pausing index from left to right (calculated using untreated wild-type data from combined replicates). The top four panels represent wild-type data, while the bottom four panels represent *cdk9^as^* data. Each quartile contains 841 filtered genes. **c** Cumulative density functions for pausing index (log₁₀) of all filtered genes in treated (red) and untreated (blue) samples for the *cdk9^as^* strain. Data used in **b** and **c** reflect the results of combined data from two biological replicates for each treatment

5 min of *cdk9^as^* inhibition, pausing did not explain this effect (Fig. 1b).

Inhibition of pause escape in mammals with drugs that target *Cdk9* is known to result in an increase in engaged, promoter-proximal Pol II coinciding with loss of signal from the gene body[28,41], yielding a greater PI for many genes. In contrast, we find few genes with significant increases in promoter-proximal Pol II upon *Cdk9* inhibition (Supplementary Fig. 2c, d). Moreover, compared to wt, *cdk9^as^* cells exhibit a global decrease in PI as a result of 5 min of treatment with 3-MB-PP1 (Fig. 1c, Supplementary Fig. 2e). Together, our results reveal that *Cdk9* activity in *S. pombe* is not required for release of promoter-proximal Pol II into elongation, but it is nonetheless critical for efficient elongation across gene bodies.

**Mcs6 and Cdk9 impact Pol II at 5′ and 3′ ends of genes.** CTD modification by transcription-coupled kinases, *Mcs6* and *Lsk1*[15], can modulate the association of auxiliary components with Pol II, to facilitate co-transcriptional RNA processing[2] and elongation through chromatin[42]. Thus, in addition to *Cdk9*, we set out to measure the impact of *Mcs6* and *Lsk1* on global transcription. A novel AS variant of *Mcs6* (here called *mcs6^as5^*) was as sensitive to treatment with 3-MB-PP1 as *cdk9^as^* (Supplementary Fig. 3a, b). Importantly, no gross transcriptional differences were observed in untreated *mcs6^as5^* compared with wt (Supplementary Fig. 2a). Within minutes of treatment, we observed measurable losses in Ser5-P and Ser7-P in the *mcs6^as5^* strain, while these marks in wt cells were unaffected (Fig. 2a). For consistency, PRO-seq experiments with *mcs6^as5^* were performed using 10 μM 3-MB-PP1, which reduces Ser5-P levels to the same extent as 20 μM of the inhibitor (Supplementary Fig. 3b). The AS *Lsk1* (*lsk1^as^*) was also inhibited by 10 μM 3-MB-PP1 (Supplementary Fig. 3c). However, despite the clear impact on Ser2 phosphorylation (Supplementary Fig. 4a), there was almost no visible change in transcription upon treatment of *lsk1^as^* with 10 μM 3-MB-PP1 for 5 min (Supplementary Fig. 4b–d). In contrast, loss of *Mcs6* or *Cdk9* activity produced abrupt changes at both ends of individual genes (Fig. 2b). Most notably in *cdk9^as^*, Pol II signal appeared to decrease with increasing distance from the transcription start sites (TSS) after inhibition. We also profiled distributions of Pol II in strains containing combinations of AS kinases. Combined inhibition of either *lsk1^as^* and *cdk9^as^*, or *mcs6^as5^* and *cdk9^as^* produced transcriptional defects resembling those of *cdk9^as^* or *mcs6^as5^*

alone, although possible basal effects of the combined mutations limited further interpretation from these strains (Supplementary Fig. 5).

Composite PRO-seq profiles around TSS and cleavage and polyadenylation sites (CPS) revealed global kinase-dependent effects at both ends of transcription units (Supplementary Fig. 5). Surprisingly, the loss of signal at 3′ ends of genes upon inhibition of *Mcs6* or *Cdk9* was only observed on longer genes (Fig. 2c, genes near the bottom of heatmaps). Differential expression analysis[43] was used to verify differences in treated and untreated PRO-seq counts within three discrete regions of each gene: (1) the promoter-proximal region (TSS to +100), (2) the early gene body (+150 to +450 from the TSS), and (3) the late gene body (−300 to CPS) (Fig. 2d). Most strikingly in *cdk9^as^*, and to a lesser extent in *mcs6^as5^*, when genes were sorted by increasing length, distance from the TSS appeared to dictate the pattern of changes within the gene body regions. Although promoter-proximal regions were only subtly affected, early gene bodies were consistently found to show increases in signal in response to *Cdk9* or *Mcs6* inhibition. However, late gene bodies displayed differing results depending on the region's distance from TSS, such that short genes exhibited increased signal and long genes had decreased signal within the last 300 bp of the transcription unit (Fig. 2e).

Several mechanisms might explain the observed gene length-dependent effects on transcription. Consistent with a loss of signal from the 3′ ends of longer genes (within bottom half of heatmaps in Fig. 2c), in the absence of *cdk9^as^* activity, Pol II may exhibit a compromised processivity, preventing transcription elongation beyond a certain distance from the TSS[37]. Alternatively, the altered transcription profiles may reflect a reduced rate of early transcribing ECs incapable, or delayed in, accelerating to the natural productive speed. Assuming an unchanged initiation rate, the increased density of elongating Pol II at the 5′ end would be compatible with a reduced elongation rate of Pol II, which within 5 min may not have allowed Pol II to transcribe fully across the longest genes.

**A checkpoint during early elongation impacts Pol II rates.** We reasoned that these proposed explanations—loss of processivity leading to premature termination vs. reduced elongation rate—could be resolved by examining a time course of Pol II distribution on genes following rapid *Cdk9* inactivation. For

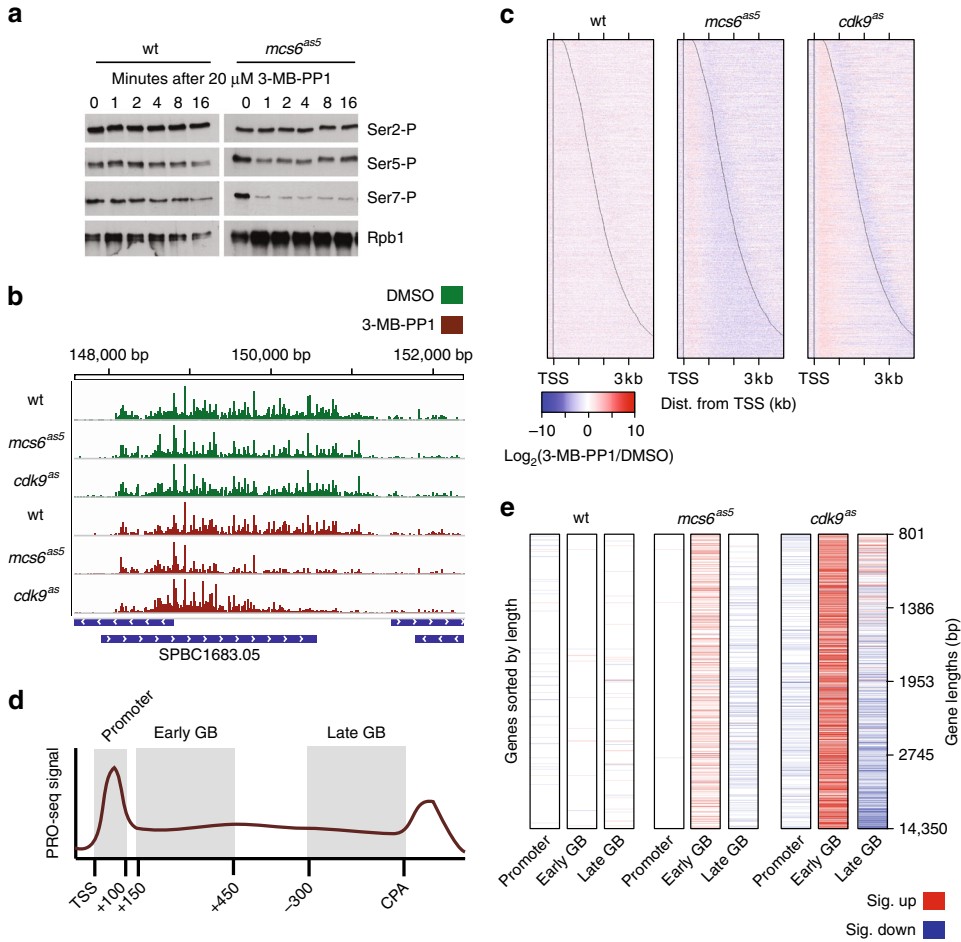

**Fig. 2** *Mcs6* and *Cdk9* impact Pol II at 5′ and 3′ ends of genes. **a** Western blot analysis of phosphorylated CTD residues (Ser2-P, Ser-5-P, Ser7-P) in relation to total CTD signal. Levels were measured over a time course of cultures treated with 20 μM 3-MB-PP1. **b** Browser track image from the *SPBC1683.05* locus displaying normalized read counts from untreated (green) and treated data (red) for wild-type, *mcs6^as5^*, and *cdk9^as^* strains from top to bottom, respectively (plus strand only). All PRO-seq samples represent treatments for 5 min with 10 μM 3-MB-PP1 (treated) or equivalent volume of DMSO (untreated). **c** Heat maps depicting the log₂ fold change in normalized PRO-seq signal (treated/untreated) within 10 bp bins from −250 bp to +4000 bp relative to the TSS ("+" indicates downstream and "−" indicates upstream) for wild-type, *mcs6^as5^*, and *cdk9^as^* strains from left to right, respectively. Data used in **b** and **c** reflect the results of combined data from two biological replicates for each treatment. Only one replicate was used for 3-MB-PP1-treated *mcs6^as5^* (see Methods). Genes within each heat map are sorted by increasing length from top to bottom, with black lines representing observed TSS and CPS (*n* = 3383). **d** Hypothetical plot for the illustration of the definition of promoter (TSS to +100) early (+150 to +450 from TSS) and late (−300 to 0 from CPS) gene body regions used in **e**. **e** Heat maps depicting whether each gene exhibits a significant fold change (adjusted *p* < 0.01; treated/untreated; DESeq2: Wald test, Benjamini and Hochberg's correction) in promoter, early, or late gene body regions. Genes were required to be longer than 800 nt (*n* = 3003) and are sorted from top to bottom by increasing gene length with length quartiles shown on the right. Significant increases and decreases in each region are shown as red and blue, respectively

instance, a defect in Pol II processivity would result in premature termination at roughly the same distance from the TSS, regardless of time after addition of 3-MB-PP1, whereas a rate defect might reveal a "wave front" shifting downstream with increasing time of inhibition. Thus, we performed a high-resolution time course of treatment with 3-MB-PP1 followed by PRO-seq, focusing specifically on the severe phenotype in *cdk9^as^*.

Strikingly, individual genes reproducibly exhibited a shifting distribution of transcribing Pol II, which appeared to propagate from the TSS toward the 3′ ends of genes over time (Fig. 3a, Supplementary Fig. 6a–c). Heat maps of the fold change after each duration of 3-MB-PP1 exposure, relative to untreated (DMSO), revealed an immediate global impact of *Cdk9* inhibition on transcription that progressed as a spreading of increased signal across genes over time. Meanwhile, an early drop in signal at gene 3′ ends was gradually recovered with the advancing Pol II density (Fig. 3b).

To better understand the propagation of increased Pol II density following the initial loss in downstream signal, we prepared composite profiles at each time-point considering only the longest genes in our filtered set (at least 6 kb, *n* = 42). Intriguingly, average profiles for these genes reveal a rapid recession of Pol II density toward the CPS (Fig. 3c)—a phenomenon also obvious within individual genes (Fig. 3a; see 1-min and 2.5-min time points).

We reason that these dynamic profiles may represent two distinct populations of transcribing Pol II captured at the moment of *cdk9^as^* inhibition. One population had yet to transit a 5′ checkpoint dependent on *Cdk9* activity and is thus impaired in elongation when *Cdk9* is inhibited. The other population, already beyond this checkpoint, may no longer require *Cdk9* to maintain its rate of transcription (Fig. 3d), which has previously been estimated in budding yeast to be ~2 kb/min[37]. Accordingly, these observations are indicative of a *Cdk9*-dependent regulatory

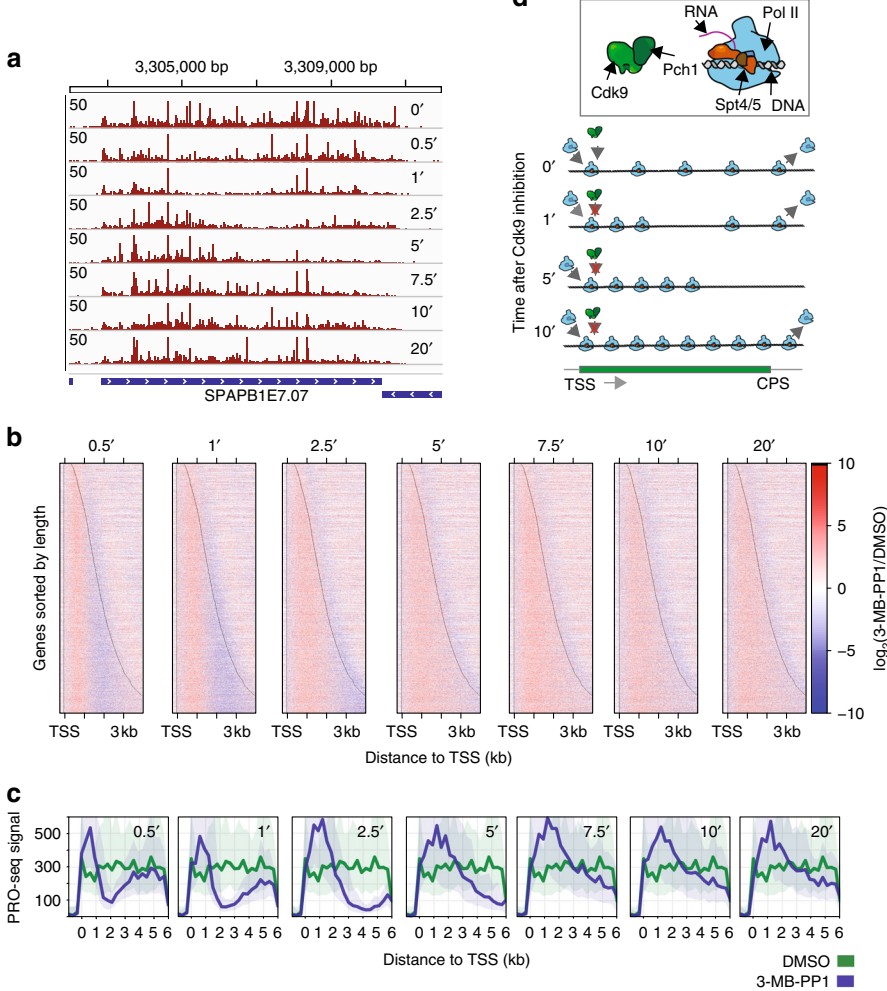

**Fig. 3** A checkpoint during early elongation impacts Pol II rates. **a** Browser track image displaying the normalized PRO-seq read count signal at the *SPBAPB1E7.07* locus for *cdk9^as* cells treated with 10 μM 3-MB-PP1 for an increasing duration, from top to bottom. **b** Heat maps of $\log_2$(treated/untreated) normalized PRO-seq signal within 10 bp windows from −250 bp to +4000 bp around the TSS for all filtered genes ($n = 3383$) ordered by increasing gene length from top to bottom. Panels show PRO-seq data from each time point of drug treatment (relative to the 0-min sample), with increasing duration from left to right. **c** Composite PRO-seq signal for all filtered genes at least 6 kb in length ($n = 42$) before and after treatment. Panels from left to right show profiles after increasing duration of treatment compared with DMSO-treated cells. Data used in **a–c** reflect the results of combined data from two biological replicates for each treatment. **d** Illustration of two populations of transcribing Pol II, which have rates differentially affected by *Cdk9* inhibition

step during the early stages of elongation in fission yeast (Fig. 3d), albeit not one fixed at a specific pause position, as is the case in mammals[29].

**Distinct effects of Cdk9 inhibition on Pol II rates**. Gene activation and repression have been used to derive transcription kinetics in mammals by following changes in Pol II distributions over time[28,44,45]. By adapting a previously described model[46], the high temporal and spatial resolution of our data allowed us to determine the distance covered by the advancing waves accurately, despite the comparatively short lengths of *S. pombe* genes (Fig. 4a). For 43 filtered genes longer than 4 kb we were able to estimate advancing wave distance at every time point from 30 s to 5 min based on the differences between treated and untreated signals within tiled windows across each gene (Supplementary Fig. 7a). Consistent with a forward moving population of Pol II, distance estimates increased with time of *Cdk9* inhibition (Fig. 4b). Moreover, by fitting a linear regression to distance traveled over time elapsed, we determined that the average rate of transcription in this population was 376 bp/min (Fig. 4c,

Supplementary Fig. 8a), much lower than previous estimates in yeast[37].

Slower moving Pol II is thought to be an easier target for exonuclease-mediated termination, as the window within which Pol II terminates downstream of the CPS is directly related to transcription rates[47]. We therefore expected that on average, the slow population of polymerase, once downstream of the CPS would terminate within a shorter distance from the CPS. Indeed, analysis of global, post-CPS elongation after 20 min of *Cdk9* inhibition shows a narrowed zone of termination (Supplementary Fig. 8b) and significantly reduced PRO-seq signal within this region when compared with untreated cells (Supplementary Fig. 8c, d).

The transient expansion of the region of Pol II clearance seen on the longest genes (Fig. 3c) suggests that Pol II further into the gene body is moving faster than the advancing wave upon loss of *Cdk9* activity. To determine the rate of this "clearing" population of Pol II, we further adapted the Danko et al.[46] model to identify the region between the advancing and clearing waves, which thus contains the start of the clearing wave (Supplementary Fig. 7b). Because the clearing population is fleeting and only present on

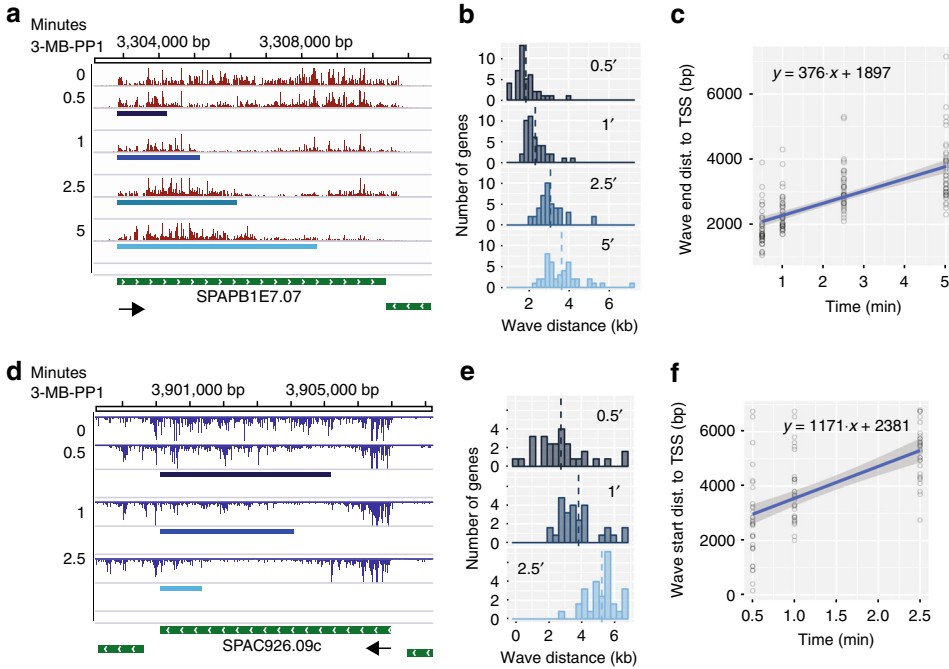

**Fig. 4** Distinct effects of Cdk9 inhibition on Pol II rates. **a** Browser image displaying normalized PRO-seq signal reflecting combined data from two biological replicates for increasing lengths of *cdk9^as* inhibition (top to bottom) over the *SPBAPB1E7.07* locus with advancing wave estimates (blue rectangles) for each time point (0.5, 1, 2.5, and 5 min) plotted below the corresponding data. **b** Histograms of estimated advancing wave end distances (relative to TSS) for filtered genes ($n = 43$) at each time point after addition of 10 μM 3-MB-PP1 to *cdk9^as* cells. Dotted, vertical lines represent the mean distance for each distribution. **c** Linear regression of distances traveled against time elapsed gives the estimated rate of advancing wave progression (slope). **d** Browser image showing clearing wave measurements (blue rectangles) for each time point plotted below the corresponding data at the *SPAC926.09c* locus. **e** Histograms of estimated clearing wave tail end distances (relative to TSS) for filtered genes ($n = 32$) at time points (0.5, 1, and 2.5 min) after *cdk9^as* inhibition. **f** Linear regression of clearing wave distances traveled vs. time elapsed gives the estimated rate of clearing wave progression (slope)

the longest genes, we were forced to omit the 5-min time point and further restrict our analysis to 32 genes longer than 6 kb. Again, clearing wave positions predicted by the model agreed well with PRO-seq profiles (Fig. 4d) and receded away from the TSS on average (Fig. 4e). Moreover, the average transcription rate of clearing complexes was estimated to be 1171 bp/min (Fig. 4f), significantly faster than the advancing wave (Supplementary Fig. 8a). This rapid clearing of gene bodies is reminiscent of the effects of FP in mammals, where late transcribing polymerases continue rapidly to clear off genes, apparently unaffected by the drug[28]. Additionally, our results demonstrate disparate effects of *Cdk9* inhibition on the elongation complex, dictated by its position relative to the TSS. Similar to the inability of metazoan Pol II to escape promoter proximal pausing in the absence of *Cdk9* activity[4], we observe a general requirement for *Cdk9* at the 5′ ends of genes in *S. pombe*. These observations suggest a general, spatially localized requirement for *Cdk9* activity on the elongation complex within the promoter-proximal region that is conserved in fission yeast. In the absence of this modification, most Pol II is able to transcribe the full length of even the longest genes, but suffers a significantly reduced elongation rate.

## Discussion
Discerning how post-translational modifications of the elongation complex influence transcription is essential for understanding gene regulation, as these mechanisms are capable of tuning the output of gene expression[29]. The work presented here advances our understanding of three transcription-coupled kinases, *Cdk9*, *Mcs6*, and *Lsk1*, and their conserved roles in transcription elongation.

Unlike *Cdk9* and *Mcs6*, *Lsk1* is non-essential in *S. pombe*[48], and loss of *lsk1^as* activity had little impact on transcription elongation

within 5 min, despite clearly affecting Ser2-P levels. Nonetheless, subtle reductions in elongating Pol II seen within the 3′ post-CPS termination region (Supplementary Fig. 4b) might be a symptom of inefficient recruitment of 3′ processing machinery[15]. On the other hand, 5-min inhibition of *Cdk9*, *Mcs6*, or both, produced highly related shifts in genome-wide Pol II distributions toward the 5′ ends of genes. Interestingly, modification of Pol II by *Mcs6* has previously been suggested to enhance recruitment of *Cdk9* in *S. pombe*[34]. Consequently, the inhibition of *Mcs6* might indirectly inhibit *Cdk9*, producing a *cdk9^as*-like effect on transcription. However, experiments in budding yeast using an irreversible AS *Kin28* mutant produced altered transcription profiles similar to what we observe after 5 min in *S. pombe*, but a full hour after inhibition[35].

As previously reported, we find that the most striking effect of inhibiting *Cdk9* is a rapid reduction of *Spt5* phosphorylation[17]. This near complete loss of *Spt5* phosphorylation within minutes of *cdk9^as* inhibition is consistent with the identification of a phosphatase opposing the activity of *Cdk9* throughout the transcription cycle[40]. Genome-wide analysis of elongating Pol II within this timeframe of *Cdk9* inhibition communicates several novel mechanistic insights into the possible role for this activity in transcription elongation. *Cdk9* activity is not essential for escape of transcribing Pol II from the promoter-proximal region in *S. pombe*, but without *Cdk9* activity these transcription complexes exhibit a severely impaired elongation rate. The progression of increased PRO-seq signal (compared with untreated cells) across the entire length of genes with increasing time of *Cdk9* inhibition excludes premature termination as the sole explanation for effects on Pol II elongation. However, we cannot rule out a combination of defects in the elongation rate and processivity of affected Pol II, which might account for non-uniform increases in Pol II

density seen at later time points after $cdk9^{as}$ inhibition (see Fig. 3c).

Surprisingly, our Pol II distributions upon loss of Spt5 phosphorylation differ notably from the depletions of total Spt5 in *S. pombe*[19]. Shetty et al.[19] reported general reduction of transcribing Pol II within gene bodies after 2 h of Spt5 depletion. In contrast, continued inhibition of $cdk9^{as}$ (10–20 min) resulted in higher density across genes due to reduced elongation rate, similar to the phenotype of strains lacking the Spt5 binding partner Spt4[32]. Importantly, we cannot rule out effects being a consequence of another known or unknown Cdk9 target. Considering that Cdk9 is capable of targeting additional transcription-coupled substrates, including the Pol II largest subunit CTD[49], it is possible that the effects we observe are attributable to other targets, or combinations thereof. Alternatively, the discrepancy of effects on Pol II transcription may represent separable functions of Spt5. The depletion of Spt5 might directly influence transcription elongation and processivity through loss of known interactions with Pol II, template DNA, and the nascent RNA[18,37,50], whereas inhibiting-specific phosphorylation of the unstructured, CTR of Spt5 might only affect transcription rate, perhaps by impairing recruitment of additional factors[51].

The loss of Spt5 phosphorylation at the early stages of elongation may reduce association of factors, which help to negotiate nucleosomes[52]. Transcription elongation is known to both, modulate and be influenced by the chromatin environment[23,53,54]. For example, inability to recruit Paf1 and the facilitates chromatin transcription (FACT) complex could reduce the ability of the elongation complex to progress through nucleosomes[55]. Alternatively, Pol II slowing may occur to ensure recruitment of 5′ capping components, similar to pre-mRNA processing steps at gene 3′ ends[22,56]. Inhibiting phosphorylation of Spt5 might prevent or delay the association of capping enzymes, potentially reducing elongation rates across entire transcription units. Nonetheless, further work comparing the importance of Spt5 phosphorylation with other Cdk9 substrates will be crucial for understanding the mechanisms by which Cdk9 exerts influence over the kinetics of transcription elongation in fission yeast.

The observation that Pol II farther downstream of the TSS (~1 kb) was less, or possibly unaffected by Cdk9 inhibition appears directly comparable to clearing waves observed in metazoan systems following the inhibition of P-TEFb[28]. But whereas most other transcribing Pol II is trapped at the promoter-proximal pause site in flies and mammals[4,28], these early gene body elongation complexes continue through the gene with severely impaired rates in *S. pombe*. We propose that NELF, which is a critical pausing factor in *Drosophila* and mammals[57–59], may serve to exploit an early elongation checkpoint, observed here in fission yeast, creating a rate limiting, Cdk9-dependent, mechanism to control gene expression. Indeed, inhibiting the ability of NELF to associate with Pol II in flies has been found to reduce the effects of Cdk9 inhibition on pause intensity[60]. Further work will be required to determine if the four-subunit NELF complex is sufficient to induce Cdk9-regulated pausing in other eukaryotes, and whether reduced Pol II elongation rates would result from pause escape in the absence of both NELF and Cdk9.

## Methods

**Yeast strains**. Yeast strains are listed in Supplementary Table 1.

**Construction of $mcs6^{as5}$ mutant fission yeast**. We previously described mcs6 and cdk9 AS mutant strains, each with single substitutions of the gatekeeper residues, Leu87 and Thr120, respectively, with Gly[34]. Although growth of both strains was sensitive to 3-MB-PP1, the $mcs6^{L87G}$ strain was ~5-fold less sensitive than was the $cdk9^{T120G}$ strain[34]. By standard methods for gene replacement in *S. pombe*[61], we introduced structure-guided second site mutations to attempt to optimize

performance of $mcs6^{as}$ alleles[62] and obtained a double-point mutant, $mcs6^{N84T/L87G}$ (referred to hereafter as $mcs6^{as5}$), which had a growth rate indistinguishable from that of wild-type cells in the absence of 3-MB-PP1, and was sensitive to growth inhibition by 3-MB-PP1 with an $IC_{50} \approx 5\,\mu M$, which was nearly identical to that of $cdk9^{as}$ (Supplementary Fig. 3a).

**Monitoring loss of target phosphorylation upon kinase inhibition**. For measuring loss of phosphorylation on Spt5 and Rpb1 upon CDK inhibition, cell lysates were rapidly prepared using the trichloroacetic acid (TCA) lysis method[63]. Briefly, cultures were grown to a density of $\sim 1.2 \times 10^7$ cells/mL in YES media at 30 °C prior to treatment. After the specified duration of treatment with of 3-MB-PP1 or DMSO, $\sim 6 \times 10^7$ cells were transferred into 500 µl 100% (w/v) TCA and collected by centrifugation. Pellets were resuspended in 20% (w/v) TCA and protein extracts were prepared using a bead beater to mechanically lyse cells with glass beads. Antibodies used in this study recognized Spt5-P (used at 1:2000) or total Spt5 (used at 1:500)[64], total Pol II (BioLegend, MMS-126R; used at 1:1000), Pol II Ser2-P (Abcam, ab5095; used at 1:1000), Pol II Ser5-P (BETHYL A304-408A; used at 1:1000), Pol II Ser7-P (EMD Millipore 04-1570; used at 1:1000), and α-tubulin (Sigma, T-5168; used at 1:5000). Raw images of all scanned western blots used in this work are provided (Supplementary Figs. 9–11).

**Treatment of AS strains for PRO-seq**. All samples were grown and treated in YES medium at 30 °C. Biological replicates were prepared by picking distinct colonies of a particular strain and growing them in separate liquid cultures. For each experiment, cultures were grown overnight and diluted to an optical density $OD_{600} = 0.2$. Diluted cultures were grown to mid-log phase ($OD_{600} = 0.5$–0.6), before treatments. Prior to treatment, all culture volumes were adjusted (based on $OD_{600}$) to have an equal number of *S. pombe* cells (in 10 mL volume). After normalization for cell number, a fixed amount of thawed *S. cerevisiae* (50 µL $OD_{600} = 0.68$) culture was spiked in to each sample. Treatments were performed by adding either 10 µM 3-MB-PP1 (stock concentration = 40 mM) or an equivalent volume of DMSO. Sample treatments were terminated by pouring into 30 mL ice-cold water, treated with diethyl pyrocarbonate DEPC. Samples were immediately spun down and processed for cell permeabilization as described below (see PRO-seq section).

**Time course experiments**. The time course experiment was performed in two biological replicates, where all time points for a given replicate was performed on cells from the same large culture. Each time point treatment was performed as described above. In order to limit differences in time on ice before permeabilization, treatment time points were carried out in reverse order. The 2.5, 1, and 0.5-min treatments were performed separately to ensure accuracy of timing. The 0-min time point was treated with DMSO for 20 min to account for any possible effects of adding the solvent for the maximum duration.

**PRO-seq**. After experimental treatments and spiking in *S. cerevisiae*, cells were spun down and washed in cold $H_2 0$. Cells were then permeabilized in 10 mL 0.5% sarkosyl at 4 °C, and incubated on ice for 20 min, then spun at a reduced RCF (400 × g) for 5 min at 4 °C. For run-on reactions, yeast pellets were resuspended in 120 µL of 2.5× run-on reaction buffer [50 mM Tris-HCl, pH 7.7, 500 mM KCL, 12.5 mM MgCl$_2$] with 6 µL 0.1 M DTT and 3.75 µL of each 1 mM biotin-NTP added immediately before use. The volume of the run-on reaction mix was brought to 285 µL with DEPC-treated $H_2 0$. 15 µL 10% sarkosyl was added to the reaction and the run-on was allowed to run on for 5 min at 30 °C. RNA was extracted using a hot phenol approach[65]; after the run-on reaction cells were pelleted at 400 × g for 5 min at 4 °C and quickly resuspended in 500 µL acid phenol. An equal volume of AES buffer [50 mM NaAc, pH 5.3, 10 mM EDTA, 1% SDS] was added and placed at 65 °C for 5 min with periodic vortexing, followed by 5 min on ice. 200 µL chloroform was added and mixed followed by centrifugation at 14,000 × g for 5 min (4 °C). 3 M NaOAc was added to the aqueous layer (200 mM) followed by ethanol precipitation with 3× volume of 100% ethanol. The RNA pellet was air dried before being resuspended in 20 µL DEPC-treated water. The standard PRO-seq protocol[66] was then followed, beginning with base hydrolysis, through to sequencing. For base hydrolysis of RNA, 5 µL 1 N NaOH was added to the resuspended RNA and left on ice for 10 min before being neutralized with 25 µL 1 M Tris-HCL pH 6.8. After passing hydrolyzed RNA through a P-30 column (Bio-Rad), RNA was incubated with streptavidin-conjugated magnetic beads (M280, Invitrogen) for 20 min at room temperature, then washed with 500 µL of each of the following: high salt solution (50 mM Tris-Cl, pH 7.4, 2 M NaCl, 0.5% Triton X-100), bead binding solution (10 mM Tris-Cl, pH 7.4, 300 mM NaCl, 0.1% Triton X-100), low salt solution (5 mM Tris-Cl, pH 7.4, 0.1% Triton X-100). Note that prior to incubation with sample RNA, M280 beads were washed once with a solution of 50 mM NaCl and 0.1 N NaOH, and then twice with 100 mM NaCl. After bead binding and washing, enriched nascent RNA was extracted using TRIzol (Ambion) and ethanol precipitated. An RNA adapter was then ligated to the 3′ ends of enriched RNAs (see below) using T4 RNA ligase 1 (NEB), followed by another round of bead binding, washing, and TRIzol extraction, as before. RNA 5′ ends were then biochemically prepared for adapter ligation by first removing any cap structures with RNA 5′ pyrophosphohydrolase (RppH, NEB), then restoring mono-phosphates to base hydrolyzed RNA 5′ ends with T4 polynucleotide kinase

(T4 PNK, NEB). T4 RNA ligase 1 was then used to ligate the 5′ RNA adapter (5′-CCUUGGCACCCGAGAAUUCCA-3′) to enriched RNAs. Following another round of bead binding and washing, as before, cDNA was prepared by reverse transcription using superscript reverse transcriptase III (SSRTIII, Invitrogen) and the primer, RP1 (Illumina, TruSeq Small RNA Sample Prep oligos). Libraries were then amplified via PCR using Phusion (NEB) polymerase and Illumina oligos, RP1 and one indexed RNA PCR Primer (RPI-x; Illumina, TruSeq Small RNA Sample Prep oligos). PCR products were run on a native polyacrylamide gel, size selected (~130–400 bp), extracted, quantified, and submitted to sequencing. More details regarding preparing PRO-seq libraries can be found in the published protocol for precision run-on and sequencing[66].

For specific batches of experiments slight modifications to the library preparation were made. Samples prepared with wt, $mcs6^{as5}$, or $cdk9^{as}$, excluding the time course experiment, were prepared according to the standard procedure described above[66]. All libraries prepared from $lsk1^{as}$ mutant and $mcs6^{as5}$ $cdk9^{as}$ double mutant strains received a novel 3′ RNA adapter during the first RNA ligation step. This 3′ adapter contains a random 6 nt unique molecular identifier at the 5′ end that, although not used here, can be used during read processing to remove PCR duplicates (5′-/5Phos/ NNNNNNGAUCGUCGGACUGUAGAACUCUGAAC/Inverted dT/-3′). For all time course samples prepared in $cdk9^{as}$, as well as $lsk1^{as}$ $cdk9^{as}$ samples, a different 3′ RNA adapter was used. Here, the RNA oligonucleotide possessed a known hexanucleotide sequence preceded by a guanine at the 5′ end, which differed for each library prepared (5′-/5Phos/ GNNNNNNGAUCGUCGGACUGUAGAACUCUGAAC/Inverted dT/-3′). The ligation of this adapter with distinct barcodes to each library permitted the pooling of all libraries after the 3′ end ligation step and thus facilitated handling. After pooling, the remainder of the library preparation was carried out as described above, but in a single tube. After sequencing, the inline barcode was used to parse reads based on their sample of origin.

**Alignment and data processing**. All sequencing was performed on an Illumina NextSeq 500 device. If samples were pooled during the library preparation using the novel 3′ RNA adapter described above, raw reads were parsed into their respective samples based on the inline barcode using fastx_barcode_splitter function from the FASTX-toolkit (http://hannonlab.cshl.edu/fastx_toolkit/). Raw reads from each sample were processed by removing any instances of partial or complete matches to the 5′ adapter sequence (5′-TGGAATTCTCGGGTGCCAAGG-3′) with the fastx_clipper function. Next, if a 3′ molecular, or sample barcode was included during 3′ adapter ligation, this length was removed from the beginning of each read. All reads were then trimmed to a maximum remaining length of 36 nt using the fastx_trimmer function. With the fastx_reverse_complement function we retrieved the reverse complement sequence of each read. All downstream alignments were performed using Bowtie (version 1.0.0)[67]. Reverse complemented reads derived from ribosomal RNA genes were then removed through alignment to a genome consisting of ribosomal RNA genes from *Saccharomyces cereivisiae* (spike-in) and then *S. pombe*. To parse reads based on organism of origin, while eliminating reads of ambiguous origin, non-ribosomal DNA reads were then aligned to a combined genome, containing each chromosome from *S. cerevisiae* (sacCer3 = S288C_reference_genome_R64-1-1_20110203) and *S. pombe* (version: ASM294v2). Only uniquely mapping reads were considered for downstream analysis. Ultimately, the normalization factor for each library was calculated as: total spike-in mapped reads/$10^5$. Unique read alignments to the fission yeast genome were processed using the Bedtools[68] function, genomeCoverageBed, to generate bedgraph formatted files, reporting the number of read 3′ ends (the last base incorporated) at each position across the genome. Bedgraph files were further converted to bigwig format for downstream analysis. Sequencing, alignment, and batch information for each sample can be found in Supplementary Table 2.

**Combined replicates**. Biological replicates correlated very well with spike-in normalization centering gene-by-gene scatter around the line $x = y$ (Supplementary Fig. 2), allowing us to combine the raw data from replicates (pre-alignment) for added read depth when useful. In one particular sample, $mcs6^{as5}$ treated with 3-MB-PP1 for 5 min, we ultimately omitted biological replicate one (Supplementary Table 2) from all analysis in this work, including composite profiles, due to concerns with the library quality. Importantly, limiting differential expression analysis between treated and untreated samples to only one replicate in the treated condition reduces power to detect differences due to overestimates of variance[43]. Although the number of real changes in the tested gene regions after inhibition of $mcs6^{as}$ may be higher than listed in Fig. 2e, the reported global shifts in Pol II distribution can be seen in individual replicates.

**Experimental batches**. Three "batches" of experiments were performed to generate the data in this work (Supplementary Table 2). In order to minimize noise introduced from across-batch comparisons, all analyses presented herein were restricted to PRO-seq libraries prepared within the same batch.

**Gene sets**. Using previously published PRO-cap data[32], we re-annotated TSS as the position with the maximum amount of background-subtracted PRO-cap signal within the region −250 to +250 around existing annotations (version: ASM294v1.16). The selected base was required to have >4 reads over background, while accounting for library size (annotations available at https://github.com/gregtbooth/Pombe_PROseq). All genes used in this work were required to have an observed TSS ($n = 4654$), and this observed TSS was used throughout. We then filtered genes for "activity" in our untreated, wild-type data. A gene was considered active if the read density within the gene body was significantly higher ($p < 0.01$) than a set of intergenic "background" regions, based on a Poisson distribution with $\lambda =$ background density ($\lambda = 0.0402$; $n$ active = 4576). Finally, to minimize the possible influence of read-through transcription from upstream genes, we filtered genes based on the relative amounts of PRO-seq signal directly upstream of the TSS. Thus, we only considered genes with more downstream (+250 to +550) relative to immediately upstream (−300 to observed TSS; $n = 3383$).

**Differential expression analysis**. Raw reads from the appropriate strand were counted within specified windows using custom scripts. Differential expression analysis was performed for desired regions using the DESeq2 R package[43]. Rather than using default between-sample normalization approaches, we supplied our own spike-in-based normalization factors (Supplementary Table 2) using the "sizeFactors" argument. Genes and regions were considered significantly changed (up or down) if they were computed to have an adjusted $p$-value < 0.01. To compute adjusted $p$-values for changes in counts DESeq2 uses the Wald test with Benjamini and Hochberg's correction[43].

**Advancing and clearing wave analysis**. Both advancing and clearing waves were identified using a three state Hidden Markov Model (HMM) that was previously developed and implemented on GRO-seq data from a human cell line[46]. To increase the number of time points for which we could identify advancing waves, we restricted our analysis to calling waves on filtered genes longer than 4 kb ($n = 231$) at each time point up to 5 min. Wave calling parameters were adjusted to accommodate the smaller genes of yeast. For calling advancing waves, the upstream region used was set to −500 to the observed TSS for each gene. Approximate wave distances, used to initialize the model, were set based on estimates derived from inspecting the fold-change heat maps (30 s = 1 kb; 1 min = 1.5 kb; 2.5 min = 2.5 kb; 5 min = 3.5 kb). Wave distances were determined based on the difference in signal within 50-bp windows between treated and untreated. The wave quality was determined based on the criteria used previously[46]. We further required that for each gene, a wave must be called for all used time points (maximum 5 min) using the combined replicate data, and that the calls must not recede toward the TSS with time ($n = 43$).

To identify regions containing the faster population of Pol II (i.e., the clearing wave), we adjusted the HMM to identify the region between the advancing and clearing waves. For this task, we used the first 1000 nt downstream of the observed TSS (within the advancing wave) to initialize the first state of the HMM. Approximate distances of the start of the clearing wave from the TSS were used to initialize the model and were set based on estimates derived from inspecting the fold-change heat maps (30 s = 3 kb; 1 min = 4 kb; 2.5 min = 5.5 kb). Unlike the model used to call the advancing wave, which assumed the upstream state to be normally distributed, all three states were presumed to behave according to distinct gamma distributions. Because of the fleeting nature of this population of polymerases observed in our time course, we were further limited to analysis of filtered genes longer than 6 kb ($n = 42$). Again, 50-nt windows were used to tile across genes; however, an additional smoothing parameter in the model arguments (TSmooth) was set to 5 as a way of restricting the effect of windows with outlying differences between treated and untreated. For each gene, we required that the model be able to identify a distinct clearing wave (Kullback–Leibler divergence > 1, between states 2 and 3) at each time point that did not move toward the TSS with time ($n = 32$).

We employed two methods to assess the confidence in measured average rates. For the first, bootstrapping approach, an average rate was calculated using 10% of genes for which we called waves. This process was repeated 1000 times (with replacement), giving a distribution of means. A more conservative, gene-by-gene estimate of variance was derived simply from the distribution of rates based on individual measurements for each gene.

**Code availability**. Custom scripts and alignment pipelines have been made publicly available through the following GitHub repository; https://github.com/gregtbooth/Pombe_PROseq.

**Data availability**. The raw and processed sequencing files have been submitted to the NCBI Gene Expression Omnibus under accession GSE102308.

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

## Acknowledgements

Research reported in this publication was supported by NIH grant GM104291 to R.P.F. and GM25232 to J.T.L. The content is solely the responsibility of the authors and does not necessarily represent the official views of the National Institutes of Health. We thank Cornell Genomics Facility for overseeing the sequencing of our libraries. We thank Chao Zhang (Department of Chemistry, University of Southern California), who originally advised us to make the N84T mutation for the generation of the *mcs6ᵃˢ⁵* strain. We are grateful for valuable discussions with Charles Danko and Hojoong Kwak regarding transcription wave calling and other computational methods.

## Author contributions

G.T.B. performed PRO-seq experiments and corresponding data analysis. P.K.P. conducted and analyzed biochemical assays. M.S. created the *mcs6ᵃˢ⁵* strain. G.T.B., P.K.P., R.P.F., and J.T.L. designed experiments and prepared the manuscript.

## Additional information

**Competing interests:** The authors declare no competing financial interests.

