## [Peer Review File · Nature Communications]

Editorial Note: This transferred paper was previous reviewed at another journal not engaging in a Transparent Peer Review scheme. This file includes the reviewer comments and author rebuttals while the paper was at Nature Communications.

REVIEWERS' COMMENTS:

Reviewer #1 (Remarks to the Author):

This is a very nice paper, and the revised version does a much clearer job explaining the new insights that can be gleaned from the Cdk9 inhibition time course of PRO-seq. In my previous review, I made a recommendation based solely on interest level, so I didn't make detailed comments. So apologize for what looks like a long list of new things. However, all my comments below are relatively minor and can be addressed without any need for re-review.

First paragraph of intro. Please add the *S. cerevisiae* names for Cdk7 and Cdk12 kinases in the introduction (as was done for Cdk9). Since many of the early papers on the CTD phosphorylation cycle uses those names, this will make it easier for readers new to the field.

Line 127 references a paper on Bioarxiv. Is it OK to cite un-reviewed pre-prints?

I had a little confusion reconciling Fig 1B showing increased PRO-seq signal in gene bodies upon cdk9 inhibition, with 2C, which shows decreases downstream. The difference turns out to be because Fig 1B only shows the first kb. Maybe this can be made clearer in the Figure 1 legend, or else a larger distance can be shown there.

Line 167 says pol II tapers off near 3' ends of genes, but the tapering isn't really linked to the 3' end. As stated much more precisely on line 182, the drop occurs beyond the first kb, independently of how much further the 3' end is.

Msc6 inactivation looks partial at best. This may be consistent with the Molina-Rodriguez paper showing that covalent inhibition may work much better.

Line 168 -172. This sentence is unclear and doesn't match well with what's in the figure. First, isn't cdk7as the same as msc6as? Second, the figure shows that the double mutants change the PRO-seq patterns even before chemical inhibition. The figure legend says the controls should limit the ability to make interpretations. But the text says the double inhibition strains resembled cdk9 and msc6 alone and therefore suggests redundancy. This sounds like an overly strong conclusion to make.

Line 239. Please provide the range of normal transcription speed estimates for *S. pombe* so the reader can judge how much slower 376 bp/min is. Presumably it's about 1 kb/min as seen for the receding wave of fast polymerase later in the paper?

Line 243. Do we really expect a narrower termination window upon Cdk9 inactivation? According to the accompanying paper, the Cdk9-dependent phosphorylation that speeds elongation is removed at the CPA site, at which point the untreated RNA pol II should also slow down.

Line 283. Given the suggestion that msc6 activity is needed for cdk9 recruitment, does inhibition of msc6 result in a drop in Spt5-P as predicted?

Line 550. Is lysis of cells in just TCA, or are glass beads also needed as for *S. cerevisiae*?

Ref 47 is missing David Bentley's name from the author list.

Reviewer #2 (Remarks to the Author):

The authors have addressed the concerns and I support the publication of the revised paper.

Minor comments/suggestions (no need for me to check this again):

- Line 45, 48, 54, 63, 87, 96: The naming of the kinases throughout the text could be more consistent, e.g. use only the gene standard names for fission yeast defined in PomBase: Msc6, Cdk9, Lsk1.

- Line 331: 'inhibition' instead of 'inhbition'.

- Line 779: ' μ M' instead of 'uM'.

- Line 1081, table: 'S. cerevisiae' instead of 'S. cereivisiae'

We congratulate the authors to a fine publication.

RESPONSE TO REVIEWERS

Reviewer #1 (Remarks to the Author):

This is a very nice paper, and the revised version does a much clearer job explaining the new insights that can be gleaned from the Cdk9 inhibition time course of PRO-seq. In my previous review, I made a recommendation based solely on interest level, so I didn't make detailed comments. So apologize for what looks like a long list of new things. However, all my comments below are relatively minor and can be addressed without any need for re-review.

First paragraph of intro. Please add the *S. cerevisiae* names for Cdk7 and Cdk12 kinases in the introduction (as was done for Cdk9). Since many of the early papers on the CTD phosphorylation cycle uses those names, this will make it easier for readers new to the field.

The *S. cerevisiae* names have been added to the introduction for each kinase.

Line 127 references a paper on Bioarxiv. Is it OK to cite un-reviewed pre-prints?

We wish to refer to this work to make the point that there is a known phosphatase that can de-phosphorylate Spt5. We will leave it to the editors of Nature Communications to decide how best to cite this work, which was originally co-submitted with this study and is now being revised for re-submission.

I had a little confusion reconciling Fig 1B showing increased PRO-seq signal in gene bodies upon cdk9 inhibition, with 2C, which shows decreases downstream. The difference turns out to be because Fig 1B only shows the first kb. Maybe this can be made clearer in the Figure 1 legend, or else a larger distance can be shown there.

The text has been modified (line 138) to specify that we are considering primarily the early gene body in this figure.

Line 167 says pol II tapers off near 3' ends of genes, but the tapering isn't really linked to the 3' end. As stated much more precisely on line 182, the drop occurs beyond the first kb, independently of how much further the 3' end is.

We understand the reviewer's concern and have amended this sentence to clarify the observation. The sentence (line 166) now reads, "Most notably in *cdk9^{Δ5}*, Pol II signal appeared to decrease with increasing distance from the TSS after inhibition."

Msc6 inactivation looks partial at best. This may be consistent with the Molina-Rodriguez paper showing that covalent inhibition may work much better.

We are unsure why the reviewer has come to this conclusion: the “kin28-is” allele developed in the Molina-Rodriguez paper relied on a second-site substitution to improve upon the “kin28-as” mutant previously published, just as our mcs6^{as5} outperforms mcs6^{as1}. Whether or not a covalent inhibitor of Msc6 would work better still cannot be inferred or predicted from these results in different species. However, we think the extended data figures 3a & b may better illustrate the extent of inhibition. We have attempted to draw further attention to these figures in the text (line 154).

Line 168 -172. This sentence is unclear and doesn't match well with what's in the figure. First, isn't cdk7as the same as msc6as? Second, the figure shows that the double mutants change the PRO-seq patterns even before chemical inhibition. The figure legend says the controls should limit the ability to make interpretations. But the text says the double inhibition strains resembled cdk9 and msc6 alone and therefore suggests redundancy. This sounds like an overly strong conclusion to make.

This was a very useful set of comments. We have adjusted the text to correct the gene names and have adjusted the text to be more cautious when considering the double mutants, due to possible basal phenotypes in these strains (lines 169-171).

Line 239. Please provide the range of normal transcription speed estimates for *S. pombe* so the reader can judge how much slower 376 bp/min is. Presumably it's about 1 kb/min as seen for the receding wave of fast polymerase later in the paper?

A reference to a previous rate estimate (Mason and Struhl 2005) has been added (lines 223-224).

Line 243. Do we really expect a narrower termination window upon Cdk9 inactivation? According to the accompanying paper, the Cdk9-dependent phosphorylation that speeds elongation is removed at the CPA site, at which point the untreated RNA pol II should also slow down.

We do expect termination to occur in a more narrow window, however we have amended our language (lines 244-245) to be more specific about slower transcription not continuing as far beyond the CPS due to facilitated termination.

Line 283. Given the suggestion that msc6 activity is needed for cdk9 recruitment, does inhibition of msc6 result in a drop in Spt5-P as predicted?

This is a good question. We have changed the text (line 284) to reflect that Msc6 action on Ser5 may enhance Cdk9 recruitment, but is not essential for Cdk9 activity toward Spt5.

Line 550. Is lysis of cells in just TCA, or are glass beads also needed as for *S. cerevisiae*?

We use glass beads and a bead beater for lysis. This has been added to the methods section.

Ref 47 is missing David Bentley's name from the author list.

This has been corrected.

Reviewer #2 (Remarks to the Author):

The authors have addressed the concerns and I support the publication of the revised paper.

Minor comments/suggestions (no need for me to check this again):

- Line 45, 48, 54, 63, 87, 96: The naming of the kinases throughout the text could be more consistent, e.g. use only the gene standard names for fission yeast defined in PomBase: Msc6, Cdk9, Lsk1.

- Line 331: 'inhibition' instead of 'inhbition'.

- Line 779: 'μM' instead of 'uM'.

- Line 1081, table: 'S. cerevisiae' instead of 'S. cereivisiae'

These corrections have been made.